# Railway Transition Curves: A Review of the State-of-the-Art and Future Research

**Tanita Fossli Brustad** *  **and Rune Dalmo**

Faculty of Engineering Science and Technology, UiT The Arctic University of Norway, Lodve Langesgate 2, 8514 Narvik, Norway; rune.dalmo@uit.no

\* Correspondence: tanita.f.brustad@uit.no

**Abstract:** Transition curves are a useful tool for lateral alignment of railway segments. Their design is important to ensure safe and comfortable travel for passengers and cargo. Well designed transition curves can lead to reduced wear of tracks and vehicles, which is beneficial from a maintenance point of view. Extensive studies have been performed through decades to find transition curves that can replace existing railway segments for the purpose of enhancing certain properties. Those studies seek to form curves that satisfy desired evaluation criteria, which are often connected to geometric continuity between the curve segments, and vehicle dynamics, to secure a smooth ride. This research topic is still ongoing and active at present. Recent results and findings are in line with the developments on the topic of vehicle dynamics and within the railway industry. For this reason it is appropriate to collect and discuss the latest work, since there are no up-to-date detailed literature reviews available. This paper explores the present state-of-the-art of railway transition curves, and identifies some of the research challenges and future research opportunities in the field.

**Keywords:** clothoid; horizontal alignment; lateral change of acceleration (LCA); railway; state-of-the-art; transition curves

---

## 1. Introduction

Most railways are described in terms of piece-wise curves. The two main types of segments, namely, straight lines and circular arcs, are connected together via transition curves. The utilization of suitable transition curves is crucial in the processes of constructing new and refurbishing existing railways. This is, in particular, important to facilitate safe and comfortable train travel, and to reduce the need for maintenance [1,2].

The main purpose of a transition curve is to enable a smooth transition between straight and curved railway segments by preventing sudden jumps in lateral forces [3], which could be the case if two main segments were coupled directly together. The advent of high speed trains and the development in heavy haul railways have triggered new requirements for transition curves. One such requirement, identified in [4], is lateral change of acceleration (LCA). The LCA function combines curve properties and vehicle parameters, to give an evaluation criterion that includes both geometry and vehicle dynamics constraints, since both are important when analysing transition curves.

Exploring new transition curves for horizontal railway alignment with more favourable properties than the classical ones is an active area of research which is highly relevant and progressing. Although the main goal of replacing one or more railway segments in both new and existing tracks is coherent in most of the research, the methods and evaluation criteria differ. Some examples of contrasts include the cover of the new curve (replacing one [5] or several [6] original segments), the applied model (simple [5] or advanced [7] vehicle model) and the principles for evaluation (lateral forces [8],

center of gravity of the vehicle [9], design simplicity [10] or cost [11]). Considerable opportunities are available to the railway industry, however, they do not come without challenges.

An important realization when it comes to optimization of railway travel and transport is that the geometric properties of the transition curve is only one part in a complex system. Transition curves alone are important, but they need to be included in a bigger picture. Research regarding safe and comfortable railways is conducted in several areas. For instance, the topics of active steering of railway vehicles [12], bogie suspension [13] and tilting trains [14] are investigated. Additionally, we note the focus on increasing the knowledge on railway dynamics [15,16], improving health monitoring systems [17,18] and further developing simulation models [19–21].

In this paper a survey of the state-of-the-art of railway transition curves is presented. The study highlights important definitions, historical events, present research, research challenges and topics for future work. The scope of the paper is to collect and discuss the latest work on railway transition curves, in order to provide an overview of research challenges and to identify relevant potential future research directions in this field. The paper focuses on the state-of-the-art of transition curves within railway applications. Nevertheless, some aspects of transition curves in highway research are also covered briefly since the two are closely connected and often referred to in the same setting.

The paper is organized as follows. Section 2 gives an overview of relevant definitions; transition curves, the classical clothoid and lateral change of acceleration, and introduces briefly the history of transition curves in railway. Section 3 presents the state-of-the-art. Challenges related to the state-of-the-art are then discussed in Section 4, and future research opportunities are considered in Section 5. Finally, some concluding remarks are given in Section 6.

## 2. Overview

This section provides some relevant definitions, to make this paper self-containing and readable for a broad audience, and covers briefly the history of the transition curve in railway.

### *2.1. Definitions*

Below follows short explanations of transition curves, the classical clothoid curve and the lateral change of acceleration concept, since these definitions and concepts are used in the sequel.

2.1.1. Transition Curves

The transition curve fulfils the role of connecting straight sections to curved sections, as well as coupling curved sections together, in railway- and highway design [22,23]. In the first case, the task of the transition curve is to gradually decrease the radius of curvature from infinity at the straight section to that of the circular curve at the curved section, and at the same time provide a change in superelevation from zero to maximum elevation. The curvature- and superelevation functions usually share the same behaviour (both are often linear), however, some research addresses the case of difference [24].

The described properties are important in order to counteract sudden jerks in the centrifugal forces, and gently apply them over the course of the transition curve. The most common transition, called simple transition curve, takes place between a straight and a curved section (see Figure 1). Transition curves are also used to enhance the change between two subsequent circular curves where the difference in radius is large. This kind is called a segmental transition curve [22], as illustrated in Figure 2. The main advantages of using transition curves are, according to [1,2], the following:

- Providing a comfortable ride for passengers.
- Providing a safer ride for passengers.
- Enabling the vehicle to drive at a higher speed.
- Reducing wear and tear on wheels and rails, decreasing maintenance and repair costs.

The clothoid is the most commonly used transition curve in road and railway design. Its curvature properties were for a long time considered to be optimal in railway alignment. However, it was

regonized later, by examining how the LCA works on the vehicle, that the clothoid is not optimal for high speed trains with regards to passenger comfort.

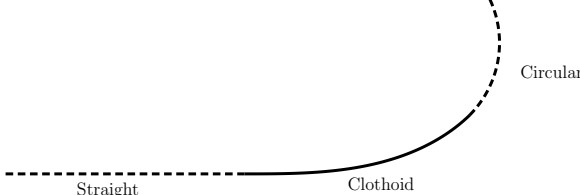

**Figure 1.** A straight curve and a circular curve connected by a transition curve.

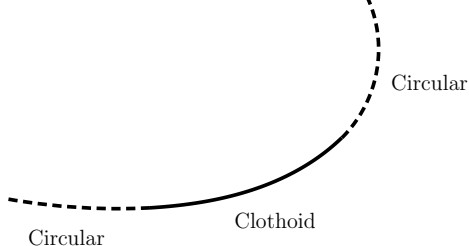

**Figure 2.** Two circular curves with different radius connected by a transition curve.

### 2.1.2. The Clothoid

The clothoid of length $l$ and end radius $r$ is a spiral defined parametrically as

$$\begin{pmatrix} x \\ y \end{pmatrix} = \begin{pmatrix} C(t) \\ S(t) \end{pmatrix}, \tag{1}$$

where $C(t)$ and $S(t)$ are the Fresnel integrals

$$
\begin{aligned}
C(t) &= \frac{1}{a} \int_0^{\hat{t}} cos\left(\frac{\pi}{2}u^2\right) du \\
S(t) &= \frac{1}{a} \int_0^{\hat{t}} sin\left(\frac{\pi}{2}u^2\right) du,
\end{aligned}
\tag{2}
$$

where $a = \sqrt{\frac{1}{\pi r l}}$ is a scaling factor, and $\hat{t} = at$, with the parameter $-\infty < t < \infty$. The following power series expansion of the integrals is given in [25]

$$
\begin{aligned}
C(t) &= \frac{1}{a} \sum_{i=0}^{\infty} \frac{(-1)^i (\frac{\pi}{2})^{2i} \hat{t}^{4i+1}}{(2i)!(4i+1)} \\
S(t) &= \frac{1}{a} \sum_{i=0}^{\infty} \frac{(-1)^i (\frac{\pi}{2})^{2i+1} \hat{t}^{4i+3}}{(2i+1)!(4i+3)}.
\end{aligned}
\tag{3}
$$

The clothoid has been proposed, solved and reinvented by a number of people throughout history. For this reason it is known by many names, such as Clothoid, Euler spiral and Cornu spiral. The history of the spiral is covered in [26]. One of the first instances of the clothoid as a transition curve in railway was by Arthur Talbot [27], who was among the first to approach the transition curve as a mathematical problem. Since then, the clothoid has become very popular and is the most widely used curve in railway geometry, mainly because of its curvature properties, which changes linearly with the curve length.

### 2.1.3. Lateral Change of Acceleration

Lateral change of acceleration (LCA) is a criterion that combines the geometry of the transition curve with variables from the vehicle to generate a vehicle dynamics measurement of the analysed curve. Baykal [4] expressed LCA as

$$z = \frac{pv}{\sqrt{u^2 + p^2}} \left( 3ka_t + v^2 \frac{dk}{dl} - \frac{kv^2u + gp}{u^2 + p^2} \frac{du}{dl} \right),$$

(4)

where $p$ is the horizontal width of the platform ($m$), $v$ is the velocity of the train ($\frac{m}{s}$), $u$ is the superelevation, which is the elevation of the outer rail according to the inner rail ($m$), $k$ is the curvature along the curve ($\frac{1}{m}$), $a_t$ is the tangential acceleration ($\frac{m}{s^2}$) and $g$ is the gravity constant (9.81 $\frac{m}{s^2}$). The LCA function describes the change in resultant acceleration occurring along the curve normal with respect to time. It can be thought of as the lateral jerk of the railway wagon. From Equation (4) it can be observed that if the velocity of the vehicle is constant, then the LCA is a scaled change in curvature. This shows that the curve geometry and the vehicle dynamics are closely connected when evaluating the suitability of a transition curve by means of LCA.

There are mainly three criteria [28] which are used in the comparison of curves with a basis in the LCA function. These are, in prioritized order,

1.  discontinuity (jumps),
2.  magnitude (extreme values) and
3.  discontinuity (breaks).

The absence of jumps in the LCA function is the most important criterion of the three. This is because discontinuities (in the form of jumps) affect travel comfort and cause wear on wheels and rails. Any curve without jumps in the LCA is considered to be superior to curves with jumps.

If neither of the curves have jumps, then they are considered to be equivalent in regards to criterion 1, and criterion 2 can be used. It compares the extreme value (the largest absolute value) of the LCA function, for each curve, against an upper boundary value. Some references for the maximum allowed value are listed in [29]:

-   $z_{max} = 0.3 \frac{m}{s^3}$ for highways [30];
-   $z_{max} = 0.3 \frac{m}{s^3}$ for railways [3];
-   $z_{max} = 0.4 \frac{m}{s^3}$ for railways [31];
-   $z_{max} = 0.6 \frac{m}{s^3}$ for highways [32];
-   $z_{max} = 0.6 \frac{m}{s^3}$ [33].

Any curve satisfying these conditions is superior to those that do not. If the curves are equivalent after criterion 2, as well, then the third criterion can be tested. Criterion 3 considers discontinuities in the form of breaks at the start and end of the transition curve. Any curve without breaks at the end-points is superior to those which has such breaks.

As noted above, the clothoid is not optimal when its LCA is examined. This can be seen from the LCA plot of a clothoid connected to a straight and circular segment depicted in Figure 3. By visual comparison of the LCA in the figure against the three criteria, it can be observed that the clothoid does not satisfy criterion 1 (since it has jumps in the graph). As consequences, passengers' comfort can be affected, in particular for high speed railways and wheels and rails can be exposed to more wear.

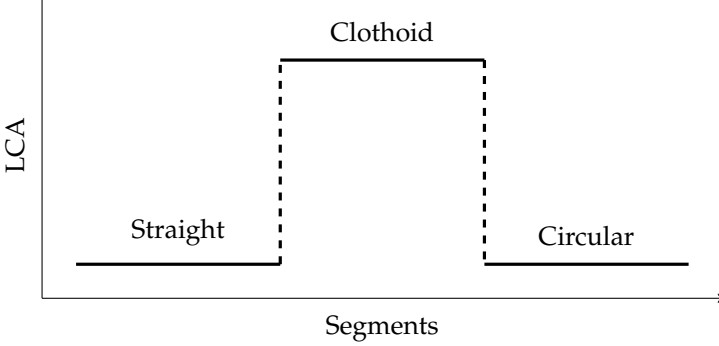

**Figure 3.** Lateral change of acceleration (LCA) plot of a clothoid connected to a straight and circular segment.

*2.2. History*

Early railway tracks used to consist of straight sections and circular arcs only. With the low train speed and the wide radius of the circular sections the direct transition between straight and circular sections was considered to be easy and safe with regards to the motion of the train [34], as well as within the scope of current surveying technologies at that time. With the evolution of railway the radius of the circular tracks decreased and the travel speed of the trains increased, and the transitions became jerky and unpleasant, and sometimes even dangerous [34,35]. This recognition paved the way for easement curves, or transition curves, which they are more known as, that enhanced the transition from a straight track to a curved track. As a result, the ride became safer and more comfortable even when higher speed was involved [34–36]. As a result of this, there are today three main track types in the horizontal layout, namely, tangent tracks (straight lines), circular tracks (with a constant radius) and transition-curve tracks [37].

Different types of transition curves have been used in railway through the years. Two early examples of such curves are addressed by prof. W. J. Macquorn Rankine in [38]: The "harmonic curve" or "curve of sines" by William Gravatt from 1829 and the "curve of adjustment" by William Froude from 1842. In the late 19th century a railway transition spiral with linear curvature, in relation to arc length, was derived independently by multiple engineers, who seemed to be unaware that Leonhard Euler had already derived that curve more than a century in prior. Consequently the curve has many names, i.e., Euler spiral, clothoid, Cornu spiral and Glover's spiral, of which clothoid is most commonly used. A. L. Higgins was the first to draw an equivalence between the railway transition spiral and Euler's definition in 1922 [39]. With emphasis on curvature, transition curves can be divided into two groups: transition curves with linear curvature and transition curves with non-linear curvature [40]. From its invention and to this day the clothoid, which belongs to the first group, is the most widely used transition curve in railway because its linear curvature property made it considered to be an optimal transition for a long time.

In addition to the clothoid, many curves with non-linear curvature were suggested as transition curves early on. Various examples are provided in [24]; the cosine curve (1868), the Helmert curve (1872), the Ruch curve (1903), the Watorek curve (1907), the Bloss curve (1936) and the sinusoidal curve (1937). Some of these curves were implemented in certain railway tracks, but they never gained the popularity of the clothoid.

The introduction of lateral change of acceleration (LCA) in 1996 [4] showed that with the increasing speeds in rail transport, the clothoid was not as optimal any more (more specifically for speeds above $120\frac{km}{h}$ [29]). This realisation opened the door to research on new types of transition curves, including parabolas [41], sinusoids [42], polynomial curves [43] and spline curves [44], which enjoy smoother LCA than the clothoid.

## 3. State-of-the-Art

This section presents the state-of-the-art research on transition curves. The presentation is assessed in two parts. The first one addresses transition curves emerging from railway research, whereas the second part deals with transition curves that origin from research within a broader range of application areas, including the combination of highway- and railway design. Some characteristics of the curves are tabulated below in Table 1.

**Table 1.** A summary of the characteristics connected to the presented transition curves.

| Curve | Characteristics | | | |
|---|---|---|---|---|
| | **Type** | **Curvature** | **Cover** | **Evaluation** |
| **Cubic parabola [45]** | Polynomial | Non-linear | Single | Curvature and jerk smoothness |
| **Sinusoid transition curves [42]** | Sinusoid | Non-linear | Single | Curvature and jerk smoothness |
| **Bloss transition curves [40]** | Bloss | Non-linear | Single | Software implementation |
| **Wiener Bogen [46]** | Curvature based | Non-linear | Single | Curvature and cant smoothness |
| **Parametric transition curves [47]** | Polynomial | Linear/non-linear hybrid | Single | Curvature smoothness |
| **Smoothed transition curves [48,49]** | Polynomial | Linear/non-linear hybrid | Single | Horizontal ordinate |
| **Optimized polynomial transition curves [43,50,51]** | Polynomial | Linear or non-linear | Single | Quality function, boundary condition |
| **General transition curves [52,53]** | Polynomial | Linear or non-linear | Multiple | Suitable LCA |
| **Universal transition curves [54,55]** | Polynomial | Linear or non-linear | Multiple | Suitable LCA |
| **Non-parametric transition curves [56]** | Bézier | Linear or non-linear | Multiple | CAD implementation |
| **Optimal layout curves [11]** | Line, arc, clothoid | Linear and constant | Multiple | Cost function |
| **Log-aesthetic curves [29]** | Log-aesthetic | Linear | Single | Suitable LCA |
| **Symmetrically Projected Transition Curves [10]** | Polynomial | Linear | Single | Simplicity, accuracy |

*3.1. Railway Transition Research*

In the recent research on transition curves in railway, the main topic revolves around finding and analyzing curves that may be suitable as transition curves. Within this topic, one subject which characterizes the work can be identified: evaluation method of curve properties. On one hand, there are simple methods involving analysis of the lateral forces that are utilized to evaluate curve properties, and on the other hand, advanced vehicle-track dynamics are exploited to evaluate the properties of the transition curve.

3.1.1. Simple Evaluation Methods

In the group that utilizes simple evaluation methods, we find work by a number of authors. The research can be divided in two main areas: research on transition curves with non-linear curvature [40,42,45,46], and research on curves possessing a linear curvature property in the middle region combined with smoothed curvatures at the ends [47–49].

In the first case, with non-linear curvature, the motivation is to improve the smoothness of lateral acceleration and lateral jerk (closely connected to LCA) in the start and end of the transition curve, when compared to linear curves. This is achieved in various ways. For example, in [45] a re-modelling of a nonlinear curvature cubic parabola is explained, where the formulation of the curve relies on a pre-evaluation of the lateral jerk. The nonlinear cubic parabola is given as

$$y(x) = \frac{x^4}{4RL^2} - \frac{x^5}{10RL^2}, \tag{5}$$

where $R$ is the radius of the circular segment, and $L$ is the length of the curve. By re-modelling the nonlinear cubic parabola in Equation (5) and considering a desired lateral jerk diagram, the proposed curve becomes the following expression

$$y(x) = \begin{cases} \frac{4x^5}{15RL^2} & \text{if } 0 \le x \le \frac{L}{4}, \\ -\frac{4x^5}{15RL^2} + \frac{2x^4}{3RL^2} - \frac{x^3}{3RL} + \frac{x^2}{12R} - \frac{Lx}{96R} + \frac{L^2}{1920R} & \text{if } \frac{L}{4} < x \le \frac{3L}{4}, \\ \frac{4x^5}{15RL^2} - \frac{2x^4}{3RL^2} + \frac{8x^3}{3RL} - \frac{13x^2}{6R} + \frac{5Lx}{6R} - \frac{L^2}{8R} & \text{if } \frac{3L}{4} < x \le L. \end{cases} \tag{6}$$

The new transition curve in Equation (6) removes the jump in lateral jerk, and smooths the transition, while keeping the maximum jerk value within a reasonable limit.

In [42] an investigation of the sinusoid as a transition curve in high speed railways is investigated. The curve formulated as

$$\Theta(l) = \frac{l^2}{2RL} + \left(\frac{L}{4\pi^2 R}\right)\left[\cos\left(\frac{2\pi l}{L}\right) - 1\right] \tag{7}$$

yields an S-shaped curvature diagram that removes the jump in lateral jerk. One reason why curves such as in Equations (6) and (7) are not widely used is because they are difficult to tabulate and stake out. Consequently, in the research, formulas are given to obtain coordinates along the sinusoid, based on the Fresnel integrals in Equation (2) from the clothoid.

Another interesting nonlinear curvature curve is the Bloss transition curve combined with a new design approach providing additional information based on Network Rail standards and international practice [40]. The Bloss curve in Equation (8) is already available in railway design software, however, important knowledge about the curve has not been implemented.

$$y(l) = \frac{1}{R}\left(\frac{l^4}{4L^2} - \frac{l^5}{10L^3}\right). \tag{8}$$

In this work, conditions regarding minimum curve length of the Bloss transition curve is derived with a basis in design rules from the Track Design Handbook [57]. The new formulas are essential towards providing accurate and good design guidance when using Bloss transitions.

Lastly, a new transition curve considering the center of gravity of the vehicle as a criterion is presented [46]. This curve is called the Wiener Bogen and it is defined with a curvature function linked to a cant function and the center of gravity,

$$\kappa(l) = \kappa_1 + (\kappa_2 - \kappa_1)f(l) - h(\psi_2 - \psi_1)\frac{d^2 f}{dl^2}, \tag{9}$$

where $\kappa_1$ and $\kappa_2$ are the curvature at each end, $\psi_1$ and $\psi_2$ are the cant values at each end, $h$ is the height to the center of gravity and $f(l)$ is the cant shaping function (chosen among 6 types, of which the most common is a seventh degree polynomial). The Wiener Bogen is implemented in Austrian railway and has proven to offer great comfort and reduced maintenance costs. Compared to the other nonlinear curvature transition curves the Wiener Bogen is unique in the way that its curvature is S-shaped (like the others), but with additional bends at the ends (which means that the curvature is not monotonically increasing). A visual explanation is given in Figure 4. A similar characteristic is not found in any of the other commonly used transition curves.

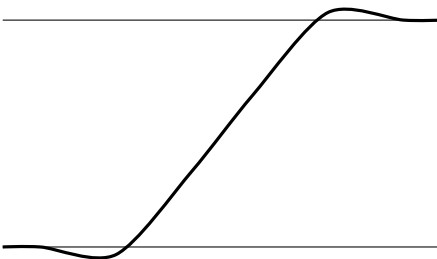

**Figure 4.** A visual explanation of the additional bends in the Wiener Bogen curvature.

All nonlinear curvature transition curves share the property that the curves are always longer than their clothoid counterparts, in order to fulfil the criteria imposed on them. This can sometimes be a disadvantage, especially in renewals of existing railway lines. However, the disadvantage of additional length is compensated by the fact that the LCA of nonlinear curvature curves tends to be of better quality, according to the three criteria give in Section 2.1.3, than the LCA of the clothoid, most notably by guaranteeing a smoother ride.

For the research on linear curvature curves with smoothed ends the motivation is similar to the case of non-linear curvature. However, the linear middle section (from the clothoid) is kept, since this part yields no problem with vehicle dynamics, and only the ends are changed. The research so far is conducted by one author, Koc, starting with a transition curve with linear curvature middle part and smoothed start/end [47], and later moving to a transition curve with linear start/middle part and smoothed end [48,49]. Both examples are transitions between a straight section and a circular section. In the research of a transition curve with smoothed start and end [47], a new curve is proposed where each region (initial region, middle region and final region) is treated separately. The purpose of the new curve is to remove the discontinuities (smooth the bends) in the curvature diagram in the initial and final regions, which is present in clothoids, to improve the dynamic properties of the rail. The parametric equations for each region are given as follows (where parameters $C$ and $D$ control the length and shape of the curves, and should be given according to Table 1 in [47]).

Initial region: $l \in [0, Cl_k]$

$$x(l) = l, \tag{10}$$

$$y(l) = \frac{A_{11}}{4}l^4 + \frac{A_{12}}{5}l^5, \tag{11}$$

where

$$A_{11} = \frac{3 - 3D(1 - C) + CD}{6C^2Rl_k^2}, \text{ and } A_{12} = -\frac{1 - D(1 - C)}{4C^3Rl_k^3}.$$

Middle region: $l \in [Cl_k, l_k - Cl_k]$

$$\begin{aligned} x(l) = & x(Cl_k) + \cos A_{20}(l - Cl_k) - \frac{1}{2}F \sin A_{20}(l - Cl_k)^2 \\ & - \frac{1}{6}(G \cos A_{20} + 2A_{23} \sin A_{20})(l - Cl_k)^3, \end{aligned} \tag{12}$$

$$\begin{aligned} y(l) = & y(Cl_k) + \sin A_{20}(l - Cl_k) + \frac{1}{2}F \cos A_{20}(l - Cl_k)^2 \\ & - \frac{1}{6}(G \sin A_{20} - 2A_{23} \cos A_{20})(l - Cl_k)^3, \end{aligned} \tag{13}$$

where $x(Cl_k)$ and $y(Cl_k)$ are from the initial region,

$$A_{20} = A_{21} + A_{22}Cl_k + A_{23}(Cl_k)^2,$$
$$A_{21} = \frac{3D - CD - 3}{12R}Cl_k, \quad A_{22} = \frac{1 - D}{2R}, \quad A_{23} = \frac{D}{2Rl_k},$$
$$F = A_{22} + 2A_{23}Cl_k, \text{ and } G = A_{22}^2 + 4A_{22}A_{23}Cl_k + 4A_{23}^2(Cl_k)^2.$$

Final region: $l \in [l_k - Cl_k, l_k]$

$$x(l) = x[(1 - C)l_k] + \cos A_{30}(l - l_{30}) - \frac{1}{2}L\sin A_{30}(l - l_{30})^2$$
$$- \frac{1}{6}(M\cos A_{30} + N\sin A_{30})(l - l_{30})^3, \tag{14}$$

$$y(l) = y[(1 - C)l_k] + \sin A_{30}(l - l_{30}) + \frac{1}{2}L\cos A_{30}(l - l_{30})^2$$
$$- \frac{1}{6}(M\sin A_{30} + N\cos A_{30})(l - l_{30})^3, \tag{15}$$

where $x[(1 - C)l_k]$ and $y[(1 - C)l_k]$ are from the middle region,

$$l_{30} = (1 - C)l_k, \quad A_{30} = A_{31} + A_{32}l_{30} + A_{33}l_{30}^2 + A_{34}l_{30}^3 + A_{35}l_{30}^4,$$
$$A_{31} = -\frac{3 - 6C + 6C^3 + 9CD - 8C^2D - 3D}{12C^3R},$$
$$A_{32} = \frac{6 - 9C + 6C^3 + 15CD - 12C^2D - 6D}{6C^3R},$$
$$A_{33} = \frac{1}{2}\frac{12 - 12C + 24CD - 16C^2D - 12D}{4C^3Rl_k},$$
$$A_{34} = \frac{1}{3}\frac{6 - 3C + CD - 4C^2D - 6D}{2C^3Rl_k^2},$$
$$A_{35} = \frac{1}{4}\frac{1 + CD - D}{C^3Rl_k^3},$$
$$L = A_{32} + 2A_{33}l_{30} + 3A_{34}l_{30}^2 + 4A_{35}l_{30}^3,$$
$$N = 2A_{33} + 6A_{34}l_{30} + 12A_{35}l_{30}^2, \text{ and}$$
$$M = 4A_{33}^2l_{30}^2 + 12A_{33}A_{34}l_{30}^3 + 16A_{33}A_{35}l_{30}^4 + 9A_{34}^2l_{30}^4$$
$$+ 24A_{34}A_{35}l_{30}^5 + 16A_{35}^2l_{30}^6 + 8A_{32}A_{35}l_{30}^3$$
$$+ 6A_{32}A_{34}l_{30}^2 + 4A_{32}A_{33}l_{30} + A_{32}^2.$$

Koc remarks in his work [47] that there should be given considerations to the occurrence of very small horizontal ordinates in the initial region (in the connection to a straight section) because of practical implementation of the transition curve. His suggestion is to reduce the length of the initial region, and choosing $C$ and $D$ accordingly. The work conducted in [48,49] further considers horizontal ordinates. This new transition curve is based on the same parametric formulas as previously described in Equations (10)–(15), however, the curve is smooth in only one end, the final region, while the remainder of the curve has a linear curvature. The smoothness in the initial region is removed with the argument that it is completely ineffective to implement and maintain in the field. Figure 5 shows approximations of the curvature behaviour for a transition curve with smoothed initial and

final regions, and a transition curve with only a smoothed final region, compared to the curvatures of
a clothoid and a Bloss curve.

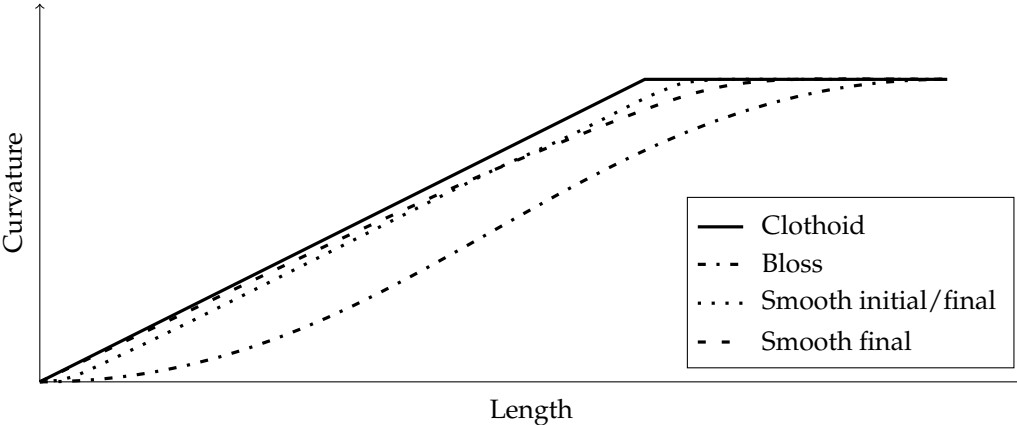

**Figure 5.** Curvature diagram for the clothoid, Bloss curve, Koc's curve with smoothed initial and final
regions and Koc's curve with smoothed final region.

### 3.1.2. Advanced Evaluation Methods

In the group that uses advanced vehicle-track dynamics the research is performed by Zboiński
together with Woźnica or Golofit-Stawinska. The two topics they focus on are forming of polynomial
transition curves with a combined use of optimization methods and advanced vehicle models [43,50,51],
and investigating the non-linear phenomena that occur in the motion of railway vehicles in transition
curves with speeds close to the vehicle's critical velocity [58]. The authors identify a change in content
of papers concerning transition curves where an attempt to get away from the standard approach of
evaluating curve properties and finding new methods is made [51]. Based on this they observe a lack of
research where advanced dynamics and optimization methods are exploited to investigate interesting
phenomena of railway vehicles in transition curves and form optimal transition shapes. The curves
chosen for optimization in [43,50,51] are polynomial curves of degree $n \geq 4$, and the objective is to
find the polynomial coefficients $A_i$, that define the curve's shape, in the following expressions (curve,
curvature, superelevation, inclination of superelevation ramp and velocity of wheel vertical rise along
the curve)

$$y(l) = \frac{1}{R}\left(\frac{A_n l^n}{L^{n-2}} + \frac{A_{n-1} l^{n-1}}{L^{n-3}} + \cdots + \frac{A_4 l^4}{L^2} + \frac{A_3 l^3}{L}\right),$$

$$\kappa(l) = \frac{1}{R}\left(n(n-1)\frac{A_n l^{n-2}}{L^{n-2}} + (n-1)(n-2)\frac{A_{n-1} l^{n-3}}{L^{n-3}} + \cdots + 3\cdot 2\frac{A_3 l}{L}\right),$$

$$h(l) = H\left(n(n-1)\frac{A_n l^{n-2}}{L^{n-2}} + (n-1)(n-2)\frac{A_{n-1} l^{n-3}}{L^{n-3}} + \cdots + 3\cdot 2\frac{A_3 l}{L}\right),$$

$$i(l) = H\left(n(n-1)(n-2)\frac{A_n l^{n-3}}{L^{n-2}} + (n-1)(n-2)(n-3)\frac{A_{n-1} l^{n-4}}{L^{n-3}}\right. \tag{16}$$

$$\left. + \cdots + 4\cdot 3\cdot 2\frac{A_4 l}{L^2} + 3\cdot 2\cdot 1\frac{A_3}{L}\right),$$

$$f(l) = Hv\left(n(n-1)(n-2)\frac{A_n l^{n-3}}{L^{n-2}} + (n-1)(n-2)(n-3)\frac{A_{n-1} l^{n-4}}{L^{n-3}}\right.$$

$$\left. + \cdots + 4\cdot 3\cdot 2\frac{A_4 l}{L^2} + 3\cdot 2\cdot 1\frac{A_3}{L}\right),$$

where $H$ is the maximum superelevation, and $v$ is the velocity. The optimization process takes into
account a model of vehicle dynamics, track-vehicle and vehicle-passenger dynamical interactions,

and a wide set of physical quantities to evaluate the curve properties, which are untypical in transition curve research. The method is still in development and many factors are not yet taken into account, including track irregularities, type of vehicle and vehicle load. However, the authors state that these factors, and more, are possible to account for in the simulation model. An advanced evaluation method offers novel opportunities in finding new transition curves, and can also reveal unknown phenomena affecting vehicles travelling through a transition curve as presented in [58]. Some results observed in the research are:

- disappearance of vibrations in the transition curve for 25TN bogie of freight cars, despite vibrations in the connected straight and circular segments;
- vibration amplitudes in the transition curve for a bogie of average parameters that exceeds amplitudes in the connected straight and circular segments;
- maximum vibration amplitude in the transition curve for a 4-axle passenger car MKIII, much higher than in the connected segments at some configurations of the suspension parameters.

*3.2. General Transition Research*

The recent research on transition curves in a general perspective (connected to both railway and highway) focus on two specific topics: replacing multiple segments (and not only the original transition curves), and finding new transition curves with better properties than the clothoid or the cubic parabola.

3.2.1. Replacing Multiple Segments

The research starts with the introduction of two new families of the general transition curves (curves with smooth and nonsmooth curvature diagrams) [52], that can be used, between two straight sections, as an alternative to the segments: first transition curve - circular arc - second transition curve, and first transition curve - second transition curve. The advantages of the new curves are stated to be the capability of describing the original segmented curve with only one equation, that the curve ensures continuous change of curvature along the entire length, and that the LCA is of better quality in the new curves compared to the clothoid. Both families are polynomial transition curves based on the function

$$y(x) = \sum_{i=0}^{n} a_i x^i, \tag{17}$$

between the start point $P$ and the end point $K$. For the family with nonsmooth curvature diagrams the polynomial in Equation (17) is of degree $n = 4$ and gives the following expression

$$y(x) = x_K(N_1 \tan u_P + N_2 \tan u_K), \tag{18}$$

where

$$N_1 = t - t^3 + \frac{1}{2}t^4, \tag{19}$$

$$N_2 = t^3 - \frac{1}{2}t^4, \tag{20}$$

and $t = \frac{x}{x_K}$, $t \in [0,1]$. For the family with smooth curvature diagrams the polynomial in Equation (17) is of degree $n = 6$ and gives the following expression

$$y(x) = x_K(F_1 \tan u_P + F_2 \tan u_K), \tag{21}$$

where

$$F_1 = t - \frac{5}{2}t^4 + 3t^5 - t^6, \tag{22}$$

$$F_2 = \frac{5}{2}t^4 - 3t^5 + t^6, \tag{23}$$

and $t = \frac{x}{x_K}$, $t \in [0, 1]$.

With a basis in the previous work a number of papers have emerged. In [54] a generalization of the general transition curves was presented as universal transition curves, UTC-1 and UTC-2, arising from Equation (17) for $n = 4$ and $n = 5$, respectively. These curves enjoy the same advantages as the general transition curves, and also offers the possibility to create curves beyond the U-shape with straight start and end, e.g., S-shaped transition curves which are useful for routing of reverse curves [55], resulting from Equation (17) for $n = 5$ and $n = 7$. Another example is the work done in [56] where Kobryń's work [6,52,54] is represented in nonparametric Bézier and nonparametric B-spline forms, making them suitable to implement into commercial software. The main goal of the work is to show that the polynomial transition curves, presented on power form, admit a straight forward representation on Bézier form as a linear combination of degree-$n$ Bernstein polynomials ($B_k^n(t)$) and $n + 1$ control points ($\{b_k\}_{k=0}^n$), see Equation (24).

$$b(t) = \sum_{k=0}^n b_k B_k^n(t), \quad B_k^n(t) = \binom{n}{k}(1-t)^{n-k}t^k, \quad t \in [0, 1]. \tag{24}$$

With geometric constraints on the control points $b_k = \{x_k, y_k\}$ the Bézier curve coincides with the graph of a degree-$n$ polynomial function in Equation (17) in Bernstein basis

$$y(x) = \sum_{k=0}^n y_k B_k^n(t), \quad t = \frac{x}{x_n} \in [0, 1]. \tag{25}$$

In this representation, few formulas describe all possible cases, and end conditions are constrained by the arrangement of the control points.

Another example of recent work on the same topic is [53] where flexible polynomial transition curves are discussed together with formulas for setting them out in the field. The two polynomial curves are the curve defined in [59] with a smooth curvature diagram

$$y(x) = \frac{x_K \tan u_P}{C}\left(Ct + \frac{2 - 5C}{2}t^4 - \frac{7 - 15C}{5}t^5 + \frac{1 - 2C}{2}t^6\right), \tag{26}$$

and the curve defined in [60] with a nonsmooth curvature diagram

$$y(x) = \frac{x_K \tan u_P}{C}\left(Ct + \frac{1 - 3C}{3}t^3 - \frac{1 - 2C}{4}t^4\right), \tag{27}$$

where $t = \frac{x}{x_K}$ for $x \in [0, x_K]$. The flexibility of Equations (26) and (27) comes from the variable

$$C = \frac{R_K \tan u_P}{x_K}, \tag{28}$$

and its permitted value options that shape the curves. In addition to the polynomial equations, formulas connected to setting out the curves in the field are defined. Among these are formulas for intermediate points $\{x', y'\}$ along the curve based on the tangent line drawn in the start point $P$,

$$x' = x \cos u_P + y \sin u_P, \tag{29}$$

and

$$y' = x \sin u_P - y \cos u_P, \tag{30}$$

with $x = tx_K$ for $t \in [0, 1]$ and $y$ is determined from curve in Equations (26) or (27).

Lastly, the work in [11] aims to give a simple and general formulation of an optimization problem to create the horizontal alignment over a longer stretch, where every geographical point has a price.

The new curve is composed of straight sections and circular arcs connected with clothoids, given by vertices $(x_i, y_i)$, radii $(R_i)$ and angles $(\omega_i)$, together with start and end points

$$x^N = (x_1, y_1, R_1, \omega_1, \ldots, x_N, y_N, R_N, \omega_N), \quad N \in \mathbb{N}. \tag{31}$$

In addition, a cost function is given, which brings together all existing costs of passing through a point $(x, y)$ in the domain. The goal is to minimise the cost function for each $N = 1, 2, \ldots$, and choose the curve with the lowest value. The cost function is described as a general function that can represent various parameters, e.g., building costs, environmental restrictions, political restrictions or terrain constraints. To test the optimization model, a section of the road joining Larraga-Lerín (NA601) in Navarra, Spain, has been considered. This particular road is of interest because its road alignment was improved through a reconstruction project. The results of NA601 compared to the curves from the optimization problem show that the curve with $N = 3$ is similar to NA601, which makes the authors believe that their proposed method can be a good way to improve alignments in road construction projects.

### 3.2.2. New Transition Curves

In [29], the family of log-aesthetic curves are examined for implementation as transition curves with a focus on revealing if they are suitable according to given criteria, and to analyse if they have a better LCA when compared to clothoids. This is the first time these curves have been studied in terms of road and railway dynamics (apart from the clothoid which is part of the log-aesthetic family), and this in itself makes them interesting. The basic formula of aesthetic curves is obtained by assuming that the logarithmic curvature histogram (LCH) is a linear function [61]

$$\log \left( \rho \frac{ds}{d\rho} \right) = \alpha \log \rho + C, \tag{32}$$

where $\rho$ is the curvature, $C$ is a constant and $\alpha$ is a slope parameter in the LCH. In the study, three log-aesthetic curves with $\alpha$ in Equation (32) set to 1, 2 and 3 are analysed in regards to smoothness criteria and LCA, and then compared to the clothoid. The results show that the log-aesthetic curves have similar dynamic properties as the clothoid, making them suitable as transition curves on the same level as clothoids.

A different approach is taken in [10] where a curve with the simplicity of the cubic parabola and the accuracy of the clothoid is introduced, the Symmetrically Projected Transition Curve (SPTC). The formula of a SPTC is as follows

$$y(x) = \frac{x^3}{2A^2} \left( \frac{1}{3} + \frac{1}{14} \left( \frac{x^2}{2A^2} \right)^2 + \frac{3}{88} \left( \frac{x^2}{2A^2} \right)^4 + \frac{1}{48} \left( \frac{x^2}{2A^2} \right)^6 + \ldots \right), \tag{33}$$

with $A^2 = RX$, where $R$ is the radius at the end of the curve, and $X$ is the curve's projection onto the x-axis. Equation (33) gives the intermediate points of the transition curve along the projection $x$. Via only considering the first term in the expression, a cubic parabola is obtained. Thus, the cubic parabola can be thought of as a first approximation of the SPTC. In the research the SPTC is compared to the clothoid and the cubic parabola, not based on vehicle-track dynamics, but rather design simplicity and accuracy, and it is thought to be a better option in cases where the cubic parabola is preferred (over the clothoid) as a transition curve.

In addition to the previous work a state-of-the-art paper has been written with focus on line to circle spiral transition curves in railway, highway, robotics, and computer imaging [62]. Challenges, in railway, highway and robotics, are noted there in the design of curves that fulfill smoothness criteria while also account for limitations in curvature, speed and obstacles, and consider energy and time efficiency. In computer imaging challenges are connected to unwanted singularities in image

interpolation when using $C^1$ or $C^2$ continuous splines. Other research challenges discussed in the same study are Hermite end conditions and shape control, and arc-length parameterization.

## 4. Challenges in Railway Transition Curve Research

Based on the state-of-the-art in Section 3, a number of research challenges can be identified. They are connected to evaluation criteria, flexibility of the new curve and linking research and industry.

### 4.1. Evaluation Criteria

From the state-of-the-art it can be observed that the evaluation criteria for evaluating new transition curves consist of both simple and more advanced methods. Analysis of curvature, lateral acceleration, lateral jerk and LCA are common approaches, and in recent research more advanced methods using vehicle-track dynamics have emerged. Although LCA is categorized as a simple method in this paper, it is more advanced than methods comparing only geometric properties of the curve, e.g., curvature. The LCA takes into account parameters from the track and the train, in addition to curve properties. However, since the method considers a railway wagon as a particle, and not a system, on the curve, it is labelled as simple.

The possibilities in evaluating curve properties in relation to its suitability as a transition curve makes it difficult to decide on an approach that is "good enough". Is an appropriate and smooth LCA enough to characterize a curve, or is a more advanced method that takes into consideration the whole system of the vehicle chain and track preferable? The question is an open one. However, it can be argued that as a first and important assessment of a new transition curve, using a simple method, e.g., LCA, is adequate, while a more advanced method may strengthen the argumentation and bring new information regarding the curve.

A strong reason for using an advanced vehicle-track model is presented in [51]. The authors state there that some dynamical effects observed in complete vehicle models are not present in models represented by a particle. Unexpected behaviour of various railway vehicles is also observed in [58] with advanced vehicle model simulations. In that study, differences can even be seen between the leading and trailing bogies when they are compared against each other for one railway car. Another reason is the knowledge that one single railway wagon most likely acts differently through a curve than a chain of wagons.

### 4.2. Flexibility of the New Curve

There are two relevant areas to consider in regards to implementing transition curves: implementing transition curves in new railway lines, and implementing transition curves in the renewal of existing railway lines. Building new railway lines implies that constrictions on the rail alignment are limited to geography and terrain challenges, which gives a larger freedom to improve the alignment when compared to renewal of old lines, where additional obstacles present in the old line must be taken into account. This means that a new transition curve has to be flexible enough to handle both of these scenarios. From the state-of-the-art research it was observed that some of the work included new transition curves that could replace either a single segment (such as the old transition curve), e.g., [10,29,46,47], or multiple segments (such as an entire turn or even more), e.g., [52,54,55].

A transition curve which replaces multiple segments may be more flexible when it comes to finding an optimal shape, but at the same time less flexible when it comes to avoiding obstacles, since a larger area of the track is involved. The opposite will be the case for transition curves replacing one segment; more flexible in avoiding obstacles, since a smaller area is involved, and less flexible in finding an optimal shape.

### 4.3. Linking Research and Industry

The link between railway transition curve research and the industry that has to implement and maintain them is sparsely covered in recent literature. Properties of a curve that are beneficial in vehicle

dynamics may not be as beneficial to implement or maintain. This can be observed in the work of Koc [48,49] where the author argues that what is considered to be an optimal curve in relation to LCA is in practice often impossible to implement and maintain because of the very low values of horizontal ordinates (and ordinates of the superelevation) in the region connected to a straight line. Koc states that implementing smooth curves in practice leads to a shortening of the transition curve, with an extension of the straight line. The conclusion is thus a recommendation to give up the condition of a smooth curvature in the end connected to a straight line, and rather keep conditions similar to those of the clothoid.

A common trend in recent papers on transition curves is to evaluate their suitability based on curve properties through analysis and simulations. Few full-scale tests, where the curve is implemented and analysed in a real world setting, have been performed. One exception is the Wiener Bogen transition curve which is implemented in Austrian railway with successful outcomes [9]. Exploring the link between simulations and full-scale experiments is relevant and important in order to fully understand the characteristics of a possible new transition curve.

Another property that should be considered is which form the curve is given on. A form that can be directly implemented into existing commercial railway software is more preferable than forms that are hard to implement. This is recognised as important in [56], where a non-parametric Bézier form in Equation (25) of the polynomial transition curves given on power form in [6,52,54] is presented. The Bézier form is commonly used in computer-aided geometric design (CAGD) and computer graphics because of its superior geometric and numerical properties [63,64] when compared to the power form. It is also simpler to enforce continuity conditions between segments for curves on Bézier form. A different approach is taken in [53] where appropriate formulas are derived, based on the curves in Equations (26) and (27), enabling calculation of data required for setting out the transition curves in the field. The formulas include tangents, normals, and coordinates, in addition to Equations (29) and (30) for calculating intermediate points $\{x', y'\}$ along the curves. Formulas for finding points along the transition curve is also presented in [42] for sinusoids. The method uses Fresnel integrals to obtain coordinates because the original Equation (7) is difficult to tabulate.

## 5. Future Research Opportunities

There are a number of interesting research opportunities connected to railway transition curves. Most of them are related to the state-of-the-art in Section 3 and the research challenges in Section 4. Here some of them are highlighted and discussed based on the authors' interests.

The problem addressed in [48,49] shows that there is a need for linking research and industry more closely. Finding curves that have beneficial properties in regards to vehicle dynamics while at the same time are possible to implement and maintain in a sustainable way can change the railway industry for the better. Future work in this area will have to include finding new evaluation criteria for the curves, from an industrial point of view, that can complement the vehicle dynamics criteria and add guidelines connected to implementation and maintenance.

The research seems to be fairly well balanced between addressing transition curves which replaces one rail segment and those dealing with multiple rail segments. However, the advantages of using one approach over the other, in existing railways, is not well documented. In theory, replacing one segment would require less work and provide more flexibility in avoiding of obstacles than what would be the case of replacing multiple segments. In practice the argumentation is not as straight forward. Replacing one segment in an already existing railway means that parts of the adjacent curves have to be shortened or extended, since the new transition curve is either longer or shorter. If the new curve, with enhanced properties, has the same length as the old segment, at least one of the adjacent segments have to be moved in order to maintain a smooth transition. This means that multiple segments will be affected either way. Research regarding the two methods with a focus on cost weighed against intervention to nature and existing railway, as well as the obtained improvements to the alignment, is a possible future work opportunity.

Although transition curve research is overflowing with families of curves suited for transition purposes, it is always interesting to analyse new curves in a transition perspective. Blending splines [65–67] are constructions where local functions at the knots are blended together by $C^k$-smooth basis functions, see Figure 6.

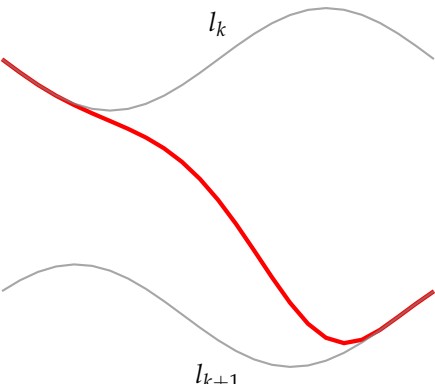

**Figure 6.** Two local curves ($l_k$ and $l_{k+1}$) blended together to create a blending spline (red curve).

They are defined in [68] as

$$f(t) = \sum_{k=1}^{n} l_k(t) B_k(t), \ t \in (t_1, t_n],$$ (34)

where $l_k(t)$ are scalar-, vector- or point-valued local functions defined on $(t_{k-1}, t_{k+1})$, $t = \{t_k\}_{k=0}^{n+1}$ is an increasing knot vector, and $B_k(t)$ are the blending functions (B-functions). Blending splines may be suitable as transition curves because of their flexibility in the blending process, connected to the choice of local functions, B-function and number of knot intervals. The curves can provide a decided degree of smoothness (which is linked to the B-function and local curves) in the knots and can be used as a replacement for one or several segments. Some initial experiments have been conducted, by the authors of this paper, which shows promising results regarding curvature smoothness and the flexibility of the blending spline. In Figure 7 the curvature of a blending spline constructed by blending two Bézier curves with an S-shaped blending function can be seen.

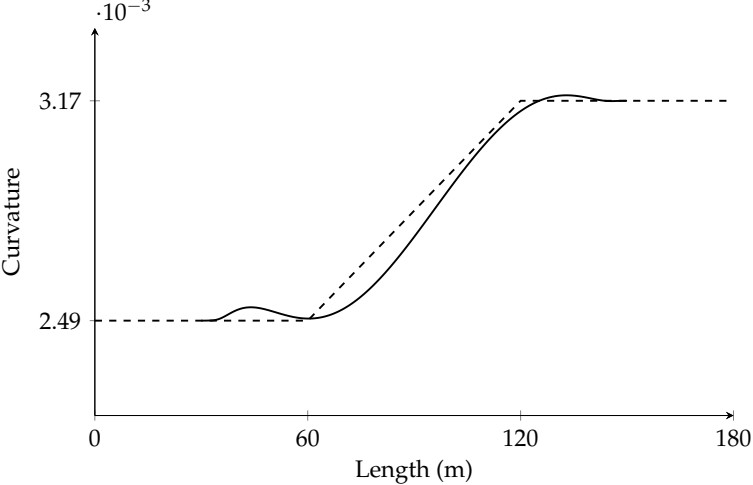

**Figure 7.** Curvature plots of a clothoid with parts of connected circular segments (dashed) and a blending spline, with local Bézier curves (solid).

The curve is placed between two circular segments, of different radii, in an existing railway, as a substitute for a clothoid. The analyses of the curvature in this blending spline transition curve show that the curvature smoothness is $G^1$ at the points of connection to the adjacent circular segments, where the clothoid is $G^0$. From the initial experiments it was observed that the curvature smoothness can be increased to $G^2$ and higher by choosing appropriate local curves, for instance by using arcs, or Bézier curves of degrees higher than 3. The research is still at an early stage, and although the end points are smooth, there remains to be put more efforts into attempts for smoothing out the overall shape of the blending spline curvature.

Another example of interesting curves are circle splines [69,70]. They are created from a blending of two circular arcs based on point positions and angles [70] as follows. Given a sequence of interpolatory constraint points $P_0, P_i, \ldots, P_i, \ldots, P_n$, a circle spline is created by blending two circular arcs ($arc_i$ and $arc_{i+1}$) for every segment ($P_i, P_{i+1}$), see Figure 8.

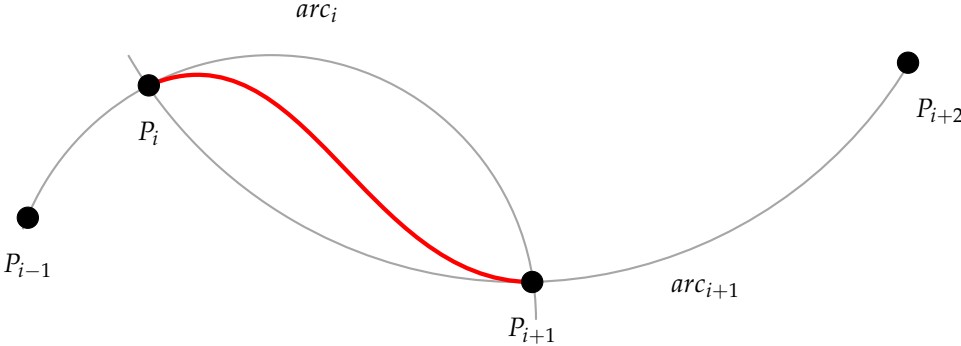

**Figure 8.** Two arcs blended together to obtain a circle spline (shown as a red curve).

Each of the two arcs are defined to go through three points, $arc_i \rightarrow P_{i-1}, P_i, P_{i+1}$ and $arc_{i+1} \rightarrow P_i, P_{i+1}, P_{i+2}$. From the arcs the tangent vectors $t_i$ and $t_{i+1}$ and the curvature of the composite curve in the points $P_i$ and $P_{i+1}$ are obtained. In the blending process, as the point $P(u)$ travels across an arc, $arc(u)$ ($u \in [0,1]$) from $P_i$ to $P_{i+1}$, the arc changes smoothly from $arc_i$ to $arc_{i+1}$. The intermediate $arc(u)$ is defined by $P_i$ and $P_{i+1}$ as well as the tangent $t(u)$ at $P_i$. To parameterize the changing process of the arcs, the directional angle

$$\tau(u) = \tau_i \cos^2\left(u\frac{\pi}{2}\right) + \tau_{i+1} \sin^2\left(u\frac{\pi}{2}\right), \tag{35}$$

created by blending $\tau_i$ and $\tau_{i+1}$ given by $t_i$ and $t_{i+1}$, is used.

A trigonometric blending function is utilized to obtain $G^2$ continuity. The result is a robustly produced fair-looking $G^2$-continuous curve without cusps or kinks. Connections between straight lines and circular arcs are also possible, where the line is considered as an arc with infinite radius. The circle spline is interesting in this setting mainly because of the way it is constructed, with arcs, and because of its smoothness.

## 6. Conclusions

In this paper, the state-of-the-art of transition curve in railway has been explored and presented, including a discussion on research challenges and future research opportunities. The paper covers essential information concerning where the research on railway transition curves is standing today. It is meant to be a source which can bring better understanding of present research challenges and inspiration for further work within the field. Transition curves in horizontal railway alignment is an important subject in relation to increase passengers' safety and reduce the need for maintenance, in particular when high speeds are involved. Geometrical properties of the curve together with vehicle dynamics can form a strong basis for the suitability of a curve as a transition curve in railway,

both for the case of designing new railways, as well as for replacing one or several segments in an existing railway.

The research on the topic is active and it is far from over. New ideas and approaches emerge in line with the developments in vehicle dynamics and the railway industry. Various branches have formed where the scientists employ different methods and criteria to solve problems in the field. Challenges in ongoing research can be observed in areas connected to evaluation criteria for a new curve, flexibility of the curve, and linking the research to the railway industry. Based on these challenges and the state-of-the-art research, future research opportunities are identified to include finding new evaluation criteria connected to the industry, research regarding advantages of replacing one or more rail segments, and investigating new families of curves, exemplified by blending splines and circle splines, as possible new transition curves.

**Author Contributions:** Conceptualization, T.F.B.; investigation, T.F.B. and R.D.; resources, T.F.B. and R.D.; writing—original draft, T.F.B.; writing—review and editing, T.F.B. and R.D. All authors have read and agreed to the published version of the manuscript.

**Funding:** This research received no external funding.

**Conflicts of Interest:** The authors declare no conflict of interest.

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
