# Peer review of "Railway Transition Curves: A Review of the State-of-the-Art and Future Research"

_infrastructures, doi:10.3390/infrastructures5050043_

Round 1

Reviewer 1 Report

The paper has a good technical content. The objectives they pursue are clear as well as the methodology used.

As a point, the reviewer proposes some changes:

- Consider that there is excessive information about what a clothoid is and what its purpose is. Such information may be appropriate for an informative article aimed at a general public, but consider that this type of scientific publication is already aimed at an expert public who knows what a clothoid is.

- The revision of the State of the Art is correct, but it would be appropriate to contribute, if it exists, information on the practical application of each commented case. For example, if it has come to use in any real way, or if I have not, what has been the reason.

- A table or graph is required where the different specific methods are visually and intuitively compared based on their main characteristics.

Author Response

The authors would like to thank the reviewer for taking the time to consider our work. We have found the feedback and suggestions to be especially helpful in the process of bringing the paper to a more final version.

Please see the attachment for a point by point answer to the comments.

Reviewer 2 Report

This paper explains about the state of the art research in railway transition curves together with an identification of research challenges and future research opportunities in the field. The title of this manuscript appears to be interesting to peers, but the content of this work looks somehow out-of-date without innovative contributions to the related (well established) methodologies. The author should make an effort to extend his idea to complex and real problems.

-Abstract: The text must be carefully revised. Some sentences contain mistakes (in the abstract: very general statements) whereas some sentences must be reworded as the English is “meaningless”. I strongly recommend that the authors retain the services of a professional editor. There are many reputable companies that offer these services.

- Introduction is poorly written. Proper references need to be used rather than using others. Language can be improved. The sentences are half constructed or incomplete in a way that the readers are expected to fend for themselves in order to understand their meaning.

I suggest the authors to review lot more papers in the following journals for more important contents, namely, Vehicle system dynamics, Multi-body simulation and others. Also, try to add a section which explains about the modelling and experimental part separately and tabulate the results. It will be very helpful for the peer researchers.

There are lot of minor errors that need to be addressed. It is difficult to mention all here. Few errors are,

(1) after equation (19), the author has not mentioned any equation number. But there are few equations represented after that in section 5.

(2) Some of the figures are well known for the railway community, so they can be removed.

(3) Many equations are represented without numbering [After equation (11)].

(4) Equations which are not relevant can be removed just be giving a proper reference.

Overall the paper appears to be of satisfactory quality. Only if the language was better, it would require lesser effort to understand and comprehend what the writer intends to convey. I would recommend this paper to be published after minor revisions.

Author Response

(The authors gave the same response as above.)

Reviewer 3 Report

The paper presents a review about the transition curves in horizontal railway.

The paper is well written and organized.

The literature review is well reported and extensively discussed.

Some formatting issues have been detected

It can be published in the present form.

Author Response

The authors would like to thank the reviewer for taking the time to consider our work. The feedback is highly appreciated.

The formatting issues may be due to the equation environment which does not work well with line numbering. This will be fixed in the final version.

Reviewer 4 Report

Dear Author,

The work is very timely as the development of High Speed Train requires innovative and reliable methods to design the infrastructure. I feel that your work needs to be refined in order to focus on the key aspects associated with the design of transition curves. The geometrical aspect per se is not enough as the vehicle dynamics and the construction implications of certain design choices are the most important factors.

I would suggest you to restructure the work taking into consideration the above points. For instance, I am not sure why you have introduced Highway transition research as the title suggests that your focus is railway. You can inform your work with the relevant points but you need to do it in a more structured way.

Author Response

(The authors gave the same response as above.)

Round 2

Reviewer 4 Report

Dear Author,

I am glad you have considered my comments and made changes accordingly.